# *Echinococcus multilocularis* Calreticulin Inhibits Lectin Pathway of Complement Activation by Directly Binding to Mannose-Binding Lectin

**DOI:** 10.3390/pathogens14040354

**Published:** 2025-04-05

**Authors:** Yuxiao Shao, Meng Xia, Yinghui Song, Yan Yan, Xiaofang Dong, Haoran Zong, Bin Zhan, Yanhai Wang, Limei Zhao

**Affiliations:** 1Department of Pathogenic Biology, School of Basic Medical Sciences and Forensic Medicine, Baotou Medical College, Baotou 014040, China; 2Department of Pediatrics, National School of Tropical Medicine, Baylor College of Medicine, Houston, TX 77030, USA; 3Parasitology Research Laboratory, School of Life Sciences, Xiamen University, Xiamen 361102, China

**Keywords:** *Echinococcus multilocularis*, calreticulin, complement lectin activation, mannose-binding lectin, immune evasion

## Abstract

Alveolar Echinococcosis (AE) is a serious zoonotic disease caused by infection of *Echinococcus multilocularis* larvae. To survive within the host, *E. multilocularis* has developed a complex immune evasion mechanism including the inhibition of complement activation. This study focused on a calreticulin secreted by *E. multilocularis* (*Em*CRT) and its role in binding ability to human MBL and inhibiting MBL-mannose-mediated lectin pathway of complement activation. Results demonstrated the binding of recombinant *Em*CRT protein to both external and natural MBL in serum and the subsequent inhibition of MBL-mannose-initiated lectin pathway reflected by the reduced formation of complement intermediate products C3b and C4b. Fragment mapping determined that the MBL binding site was located within the S-domain of *Em*CRT. Combining with its role in inhibiting C1q-initiated classical complement activation in our previous study, the inhibition of MBL-mannose-initiated lectin pathway identified in this study suggests *Em*CRT plays an important role in the immune evasion of *E. multilocularis* alveolar larvae against host complement attack as a survival strategy within human tissue. This study supports the approach of using *Em*CRT as a good candidate for vaccine and drug development against *E. multilocularis* infection.

## 1. Introduction

Echinococcosis is a severe zoonotic disease caused by infection with the larval stage of the *Echinococcus* tapeworm. There are two types of echinococcosis: cystic echinococcosis (CE) caused by infection with *Echinococcus granulosus* and alveolar echinococcosis (AE) caused by infection with *Echinococcus multilocularis* [1,2]. Both AE and CE are serious diseases. Due to infiltrative growth characteristics of the larvae of *E. multilocularis*, cases of AE in humans and animals are more serious and difficult to treat than CE. Therefore, AE is also known as “malignant echinococcosis” or “worm cancer”. Human infections with *E. multilocularis* are usually caused by accidental ingestion of parasite eggs, which develop into larvae within the human body and migrate to the liver to form foci. The infection with *E. multilocularis* takes a heavy toll on the health and well-being of humans and their domestic livestock, responsible for infecting 29,600 individuals each year and causing nearly 17,000 mortalities worldwide. Cosmopolitan distribution of this ailments had led to losses of 3 billion dollars annually [3,4,5].

Tissue-dwelling *E. multilocularis* larvae in infected humans are exposed to a hostile immune microenvironment. As a survival strategy, the parasitic larvae have developed a relatively complex immune evasion mechanism. Understanding the immune evasion mechanism of *E. multilocularis* larvae in host could facilitate the development of vaccines or drugs to control the infection of *Echinococcus* and echinococcosis.

Parasitic helminths have evolutionally developed different levels of immune evasion mechanisms; one of them is to regulate the host immune system through the secreting of functional immunomodulatory proteins [6,7]. Since the complement system is the first line of innate immunity, playing a vital role in the immune defense against pathogen infections [8], helminths develop comprehensive strategies to inhibit or block host complement recognition and activation during invasion [9,10].

The complement system contains a collection of inactive components in the blood and tissues that become activated in response to infection and initiate the immune attack to the invaded pathogens [11]. There are three complement activation pathways: classical, lectin, and alternative [12,13]. The classical pathway is triggered by antigen–antibody immune complexes binding to C1q. The lectin pathway is initiated by the binding of mannose-binding lectin (MBL) to bacterial surfaces with mannose-containing polysaccharides (mannans) leading to the formation of the MBL/MASP-1/MASP-2 tri-molecular complex and subsequent cleavage of C4 into C4a and C4b [14]. Ultimately, three activation pathways lead to the formation of membrane attack complex (MAC) that lyses and kills the pathogen [7]. In addition, MBL is also involved in the regulation of phagocytosis, mediating cytotoxicity and apoptosis, and triggering the release of pro-inflammatory factors as well as the production of reactive oxygen species ROS as other means to kill invaded pathogens [15,16].

Calreticulin (CRT) is a multifunctional protein with its structure containing a globular N-terminal, proline-rich P, and acidic C-terminal domains with different functions associated with cell adhesion, calcium storage, and phagocytosis of apoptotic cells [17]. The N and P domains play a chaperone role in the endoplasmic reticulum, while the C-domain owns Ca^2+^ binding activity [18]. It has been demonstrated that CRT was conservatively expressed in helminth parasites with a strong regulating function on the human complement system as a strategy to evade attack by a complement-initiated innate immune response [19]. Our previous study has identified tissue-dwelling nematode *Trichinella spiralis*-produced CRT (*Ts*-CRT) strongly bound to human complement component C1q to inhibit C1q-initiated complement classical activation pathway [9]. The C1q-binding CRT was also identified in *Haemonchus contortus*, *Trypanosoma cruzi*, *Necator americanus,* and *Brugia malayi* [20,21,22,23], indicating CRT is a conserved protein secreted by different helminths to defend the complement-involved host first line immune attack. Further study identified that the C1q-binding site of *Ts*-CRT was located in the S-domain between the N and P domains (135–288 aa) [10]. In our previous study, we also determined that *E. multilocularis*-derived CRT (*Em*CRT) was also able to bind on human C1q to inhibit C1q-mediated complement activation and C1q-dependent non-complement activation of immune cells [24].

Except for the C1q-binding capacity to interfere with C1q-initiated classical pathway of complement activation, CRTs produced by some other parasites such as *Trypanosoma cruzi* [25] and *T. spiralis* [26] have also been shown to be able to inhibit the activation of the lectin pathway by binding to lectin pathway activation molecules such as MBL or ficolin as an alternative approach to inhibiting complement activation in order to survive in the host.

In this study, we investigated whether *Em*CRT was able to bind to MBL to inhibit lectin pathway of complement activation as another approach to defending complement-derived innate immune attack in addition to inhibiting C1q-initiated classic activation pathway. Our results showed that *Em*CRT was able to inhibit the activation of the complement lectin pathway by binding to human MBL, and the MBL binding region was mainly located in the S domain of *Em*CRT. The results identified in this study further confirm that *E. multilocularis*-expressed CRT plays a key role in evading the host immune response and facilitating parasite survival within the host.

## 2. Materials and Methods

### 2.1. Serum

Normal human sera (NHS) were obtained from five healthy volunteers with signed consent under the protocol reviewed and approved by the Institutional Review Board (IRB) of Baotou Medical College (approval number: 2018026). Collected NHS was used as source of complement. C1q-depleted sera (C1qD) were purchased from Quidel (San Diego, CA, USA).

### 2.2. Expression of Recombinant Proteins of Full-Length EmCRT and Different Fragments

The DNA coding for the full-length *Em*CRT without signal peptide (18–395 aa) and the different domains [*Em*CRT-N (18–167 aa), *Em*CRT-P (168–325 aa), *Em*CRT-NP (18–325 aa), *Em*CRT-C (326–395 aa), and *Em*CRT-S (140–292 aa)] were originally amplified from the total cDNA of *E. multilocularis* metacestode [27] and cloned into prokaryotic expression vector pET-28a (Novagen, Darmstadt, Germany). The correctly cloned recombinant plasmids were transformed into BL21(DE3)-competent cells (TIANGEN, Beijing, China). The recombinant proteins with six His tags at N-terminus were expressed under induction of 1 mM IPTG overnight at 25 °C and purified with Ni^2+^ affinity chromatography (Beyotime Biotechnology, Shanghai, China), as described previously [24,26]. The concentration of purified recombinant proteins was determined by BCA protein concentration assay kit (Solarbio, Beijing, China).

### 2.3. EmCRT Binding to MBL

ELISA: To determine the binding ability of recombinant protein *Em*CRT (r*Em*CRT) to MBL, the 96-well plates were coated with different amounts of MBL (0, 0.5, 1, 2, 4, 8, 12, 15 μg/mL) (Sinobiological, Beijing, China) in 100 μL coating buffer (100 mM NaHCO_3_/Na_2_CO_3_, pH 9.6), and the control plates were coated with the same amount of bovine serum albumin (BSA). The wells were washed with 200 μL of 1 × PBST (1 × PBS + 0.05% Tween-20) and blocked with 200 μL of 3% BSA at 37 °C for 1 h. After another round of washing with PBST, 100 μL of binding buffer (20 mM Tris-HCl, 50 mM NaCl, 10 mM CaCl_2_, 0.05% Triton-X 100, pH 7.4) containing different amounts of r*Em*CRT (0, 0.2, 0.4, 1.2, 1.5 μM) was added to the wells for binding at 4 °C overnight. After being washed for three times, mouse anti-His-tag monoclonal antibody (BOSTER Biological Technology, Wuhan, China) at 1:5000 dilution in PBS was added into each well as the primary antibody, and HRP conjugated goat anti-mouse IgG (1:10,000, BOSTER Biological Technology, Wuhan, China) as the secondary antibody to detect the binding of r*Em*CRT to MBL. The chromogenic substrate TMB (Beyotime Biotechnology, Shanghai, China) was added, and the absorbance values were detected at OD_450_ with a microplate reader (Thermo Fisher, Waltham, MA, USA) after termination of the reaction.

Far western blot: Equal amounts of MBL and BSA (1 μg) were run SDS-PAGE on 12.5% gel and transferred to PVDF membrane (Merck, Darmstadt, Germany). After being blocked with 5% skimmed milk, the membrane was incubated with 15 μg/mL r*Em*CRT in binding buffer (20 mM Tris-HCl, 50 mM NaCl, 10 mM CaCl_2_, 0.05% Triton-X 100, pH 7.4, 1% skimmed milk) overnight at 4 °C. The binding of r*Em*CRT to MBL on the membrane was detected with mouse anti-His monoclonal antibody (1:1000, BOSTER Biological Technology, Wuhan, China) and HRP conjugated goat anti-mouse IgG (1:12,000, BOSTER Biological Technology, Wuhan, China).

To determine the specific fragment of *Em*CRT that binds to MBL, the same amounts of recombinant proteins of different fragments expressed in BL21(DE3) including r*Em*CRT-N, r*Em*CRT-P, r*Em*CRT-NP, r*Em*CRT-C, and r*Em*CRT-S were used in ELISA (0.5 µM) and Far western blot (1 µg), as described above.

### 2.4. Molecular Docking

Structural prediction of *Em*CRT 3D structure was performed using AlphaFold 3 (DeepMind, London, UK). Docking of human MBL (PDB: 1HUP) to the *Em*CRT model was performed using the professional protein–protein docking tool HDOCK v1.0 (Huazhong University of Science and Technology, Wuhan, Hubei).

### 2.5. Calcium-Binding Staining

The calcium-binding property of the purified recombinant proteins, r*Em*CRT, r*Em*CRT-N, r*Em*CRT-P, r*Em*CRT-NP, r*Em*CRT-C, and r*Em*CRT-S, were measured by staining with Stains-all (Sigma, St. Louis, MO, USA), a cationic carbocyanine dye that stains Ca^2+^-binding proteins as blue and non-Ca^2+^-binding proteins as red. BSA and non-relevant recombinant *E. multilocularis* ferritin (r*Em*Fer) (expressed in the lab) were used as non-Ca^2+^-binding protein controls.

### 2.6. Effect of rEmCRT on the Binding of MBL to Mannan

Microplates were coated with 50 μg/mL mannan (Sigma, St. Louis, MO, USA) in coating buffer at pH 9.6 overnight at 4 °C, then blocked with 3% BSA at 37 °C for 2 h. Total 1 μg of MBL was pre-incubated with different amount of r*Em*CRT or r*Em*CRT-S or r*Em*CRT-NP (0, 0.5, 1 μM) in binding buffer (20 mM Tris-HCl, 50 mM NaCl, 10 mM CaCl_2_, 0.05% Triton-X 100, pH 7.4) at 37 °C for 2 h before adding into the mannan-coated plates (100 µL). BSA (1 μM) was used as a control for the MBL binding reaction. The incubation was continued overnight at 4 °C. After washing with 1 × TBST containing 5 mM CaCl_2_, the inhibition of MBL binding to mannan was detected with mouse anti-MBL monoclonal antibody (1:3000, R&D System, Minneapolis, MN, USA) and HRP conjugated goat anti-mouse IgG (1:5000, BOSTER Biological Technology, Wuhan, China).

To mimic the natural environment of MBL in serum and exclude the C1q-involved classical complement activation pathway, C1qD serum at 1:50 dilution in binding buffer (20 mM Tris-HCl, 50 mM NaCl, 10 mM CaCl_2_, 0.05% Triton-X 100, pH 7.4) was pre-incubated with different amounts of r*Em*CRT or r*Em*CRT-S or r*Em*CRT-NP as described above. The reduced amount of MBL from C1qD serum binding to mannan on the plates was detected with anti-MBL monoclonal antibody.

### 2.7. C3b/C4b Deposition Experiments

Microplates were coated with mannan (50 μg/mL) overnight at 4 °C, then blocked with 5% BSA at 37 °C for 2 h. C1qD serum at 1:50 dilution in binding buffer (20 mM Tris-HCl, 50 mM NaCl, 10 mM CaCl_2_, 0.05% Triton-X 100, pH 7.4) was used as a natural source of MBL and pre-incubated with different amounts of r*Em*CRT or r*Em*CRT-S or r*Em*CRT-NP (0, 2, 4 μM) at 37 °C for 2 h. BSA (4 μM) was used as a non-relevant protein control. Total 100 μL of the incubation mixture was added to each well coated with mannan and incubated overnight at 4 °C. After washing with 1 × TBST containing 5 mM CaCl_2_, the C1qD diluted at 1:150 in 1 × Veronal Buffer (VB, Lonza, Basel, Switzerland) containing 0.1% gelatin, 0.05% Tween-20 was added (100 µL) as supplement of other complement components (without C1q) into each well of the plates and incubated at 37 °C for 1 h. The lectin pathway-activated C3b/C4b deposition was detected with rabbit anti-C3b monoclonal antibody (1:1000, BOSTER Biological Technology, Wuhan, China) and goat anti-C4b polyclonal antibody (1:3000, Abcam, Cambridge, UK). The HRP conjugated goat anti-rabbit IgG (1:1000, Affinity Biosciences, Liyang, China) or rabbit anti-goat IgG (1:8000, Affinity Biosciences, Liyang, China) were used as secondary antibodies, respectively.

To mimic the natural MBL in serum, after the same coating with mannan and blocking with BSA, a 1:100 dilution of NHS in binding buffer (20 mM Tris-HCl, 50 mM NaCl, 10 mM CaCl_2_, 0.05% Triton-X 100, pH 7.4) was pre-incubated with different amounts of r*Em*CRT or r*Em*CRT-S or r*Em*CRT-NP (0, 4, 8 µM) at 37 °C for 2 h before adding into plates coated with mannan at 100 μL per well for overnight at 4 °C. BSA (8 μM) was used as control for the binding. After washing with 1 × TBST containing 5 mM CaCl_2_, the other components of complement were supplemented with 1:150 with NHS in 1 × Veronal Buffer (VB, Lonza, Basel, Switzerland) containing 0.1% gelatin, 0.05% Tween-20 for incubation for 1 h at 37 °C. After washing, C3b and C4b deposition was detected in the same conditions as described above.

### 2.8. Data Analysis

All experiments were performed in duplicate and repeated at least once, and data were shown as meant ± standard deviation. The statistical analysis of the data was performed using the two-tailed Mann–Whitney test by GraphPad Prism 5 software (GraphPad Prism, Inc, San Diego, CA, USA). *p* < 0.05 was considered as statistically significant difference. We tested the data for normal distribution through the Kolmogorov–Smirnov test.

## 3. Results

### 3.1. Expression and Characterization of Recombinant EmCRT and Its Fragments

The structure of *Em*CRT and its functional domains is shown in Figure 1A based on the results from *Ts*-CRT [10]. The recombinant proteins of full-length *Em*CRT, *Em*CRT-N, *Em*CRT-P, *Em*CRT-S, and *Em*CRT-NP were successfully expressed in *E. coli* BL21 (DE3) and purified with immobilized metal affinity chromatography (IMAC). The purified recombinant proteins appeared as expected in SDS-PAGE based on the sequence size and possible molecular interaction (r*Em*CRT-N 20 kDa, r*Em*CRT-P 35 kDa, r*Em*CRT-S 35 kDa, r*Em*CRT-C 20 kDa, r*Em*CRT-NP 50 kDa, r*Em*CRT 58 kDa) and were recognized by anti-His antibody (Figure 1B) [27]. The Stains-all results revealed r*Em*CRT-P, r*Em*CRT-S, r*Em*CRT-C, r*Em*CRT-NP, and r*Em*CRT were stained blue, indicating their calcium-binding activities, while r*Em*CRT-N was stained red, indicating it does not contain calcium binding domain as control recombinant protein *Em*Fer and BSA do (Figure 1C).

### 3.2. Binding of rEmCRT to Human MBL

The binding activity of r*Em*CRT to human MBL was examined by ELISA and Far western blot. ELISA results showed that r*Em*CRT was able to bind to human MBL coated on the plates in a dose-dependent pattern (Figure 2A(a)), but did not bind to control BSA (Figure 2A(b)). SDS-PAGE revealed a single strand of MBL was present under reduced condition with molecular weight of 33 kDa (Figure 2B(a)). Far western blot with anti-His antibody showed r*Em*CRT was able to bind to MBL transferred on PVDF membrane and did not bind to the BSA (Figure 2B(b)).

### 3.3. Predicted Structure and MBL Binding Region of EmCRT

The 3D structure of *Em*CRT was predicted and constructed using AlphaFold3 prediction, and the final 3D structure of *Em*CRT was optimized and adjusted using PROCHECK (Figure 3A). The interaction between *Em*CRT and MBL was analyzed and identified that amino acid residues located within the N domain and P domain of *Em*CRT were involved in binding to the MBL A-chain (Figure 3B). Specifically, amino acids Gly^87^ and Thr^89^, in the N domain, and Arg^176^, Asp^178^, Thr^180^, Ser^194^, Lys^205^, ASP^209^, Gln^292^, Ile^293^, Pro^294^, in the P domain, and Asp^333^ in the C domain interacted with Lys^A93^, Gln^A96^, Thr^A97^, Ala^A100^, Lys^A104^, Gly^A115^, Lys^A117^, Arg^A146^, Asn^A147^, Arg^A181^, Thr^A184^, Ile^A228^ of the MBL A chain. Amino acids Glu^223^, Pro^280^, Asn^281^, Pro^282^, Tyr^284^ in the P domain interacted with Lys^C170^, Thr^C179^, Gly^C180^, Asn^C181^ of MBL C chain. Amino acid Glu^300^ in the P domain interacted with Arg^B101^ of MBL B chain. The interaction between *Em*CRT and MBL occurs mostly by hydrogen, electrostatic, and hydrophobic bonds. The docking predicted that MBL binding region in *Em*CRT is mainly located between Gly^87^ in the N-domain and Glu^300^ in the P-domain (Figure 3C). The docking results are consistent to the actual binding results shown in Figure 3.

### 3.4. Mapping of MBL Binding Region in EmCRT

The MBL binding region on *Em*CRT was determined by the fragment mapping expression and MBL binding assays. ELISA with MBL coated on the plates demonstrated that the full-length *Em*CRT as well as fragments including r*Em*CRT-P, r*Em*CRT-S, r*Em*CRT-C, r*Em*CRT-NP were able to bind to MBL in a dose-dependent manner, with r*Em*CRT-S and r*Em*CRT-NP possessing the highest MBL binding ability similar to the level of full-length r*Em*CRT conferred, which is significantly higher than that of r*Em*CRT-P and r*Em*CRT-C (Figure 4A(a)). No significant binding of r*Em*CRT-N to MBL was observed. All r*Em*CRT and its fragments did not bind to control BSA (Figure 4A(b)). Far western blot results revealed that r*Em*CRT-S and r*Em*CRT-NP appeared to bind to MBL similarly to r*Em*CRT and significantly higher than that of r*Em*CRT-N, r*Em*CRT-P, and r*Em*CRT-C (Figure 4B), which was further illustrated by densitometry analysis of recognized bands with the reading of r*Em*CRT binding band set as 1.0 (Figure 4C). The results confirm that the MBL binding region is located in r*Em*CRT-S within r*Em*CRT-NP fragment.

### 3.5. Inhibition of MBL Binding to Mannan by rEmCRT and Its Binding Regions

To determine whether the binding of r*Em*CRT and its functional regions (r*Em*CRT-S and r*Em*CRT-NP) to MBL would interfere with the binding activity of MBL to its ligand mannan, ELISA with mannan coated on the plates was performed. The ELISA results demonstrated that r*Em*CRT, as well as its functional regions r*Em*CRT-S and r*Em*CRT-NP, were able to significantly inhibit the binding of MBL to mannan in a dose-dependent manner (Figure 5A). For further validation, we used C1q-depleted sera (C1qD) to mimic the natural environment of human serum containing the physiological level of MBL. It also showed that pre-incubation r*Em*CRT, r*Em*CRT-S, and r*Em*CRT-NP with C1qD (1:50) significantly inhibited the binding of MBL in the serum onto mannan coated on the plates at the similar level as external MBL shown in Figure 5B. The results showed that r*Em*CRT and the binding regions r*Em*CRT-S and *Em*CRT-NP were able to inhibit the binding of both external and physiological MBL to mannan in a dose-dependent pattern.

### 3.6. Inhibition of Lectin Pathway of Complement Activation by rEmCRT and Its MBL Binding Regions

To identify the effect of r*Em*CRT and its MBL binding regions on the activation of the complement lectin pathway, the depositions of C3b and C4b, the intermediate product of complement activation, were tested after the lectin activation pathway was initiated. At the first experiment, we used C1qD serum as physiological source of MBL to exclude the possible classical complement activation initiated by C1q. After pre-incubation with r*Em*CRT or r*Em*CRT-S or r*Em*CRT-NP, the C1qD serum containing MBL was added to plates coated with mannan. ELISA detected much less the formation and deposition of C3b and C4b on the plates, the intermediate complement products of MBL-mannose-initiated lectin activation pathway. The *Em*CRT-induced inhibition of lectin complement activation pathway was dose-dependent (Figure 6A). For further validation, we used normal human serum (NHS) as source of MBL to mimic the human physiological environment instead of using C1qD serum. The r*Em*CRT including r*Em*CRT-S and r*Em*CRT-NP inhibited the deposition of C3b and C4b at the similar level as shown above using C1qD serum (Figure 6B), indicating the *Em*CRT-triggered inhibition of complement activation is directly derived from MBL-mannose-initiated lectin pathway, not from C1q-initiated classical pathway.

## 4. Discussion

During the million years’ parasitism in host, helminth parasites have evolutionally developed strategies to evade host immune attack by downmodulating the host immune system through secreting a spectrum of molecules with immunomodulatory functions [28,29]. Understanding the immunomodulatory mechanism that parasitic helminths bear would not only facilitate the development of vaccines or drugs against parasite infection, but also help the design of therapeutic approaches to treat diseases of immunological disorders in humans [30,31]. Currently, the research focuses more on the activation of regulatory T cells [32] and suppressive macrophage populations that might be important in downmodulating immune responses [33]. The inhibition of complement activation has been identified as another approach for helminth to live concomitantly in the host, especially for those helminths living in the tissue, such as *T. spiralis* and *E. multilocularis* [9,24].

The complement system, an important component of innate immunity, is a highly complex biological response system involved in the direct lysis and killing of invasive pathogens, promoting phagocytosis and inflammatory response, therefore a crucial component of both the innate and adaptive immune responses [7,8]. Calreticulin (CRT) is a highly conserved Ca^2+^ binding protein involved in a range of biological processes such as wound healing, protein modification and folding, secretory pathway regulation, cell motility, cellular metabolism, protein synthesis, gene expression regulation, cell cycle regulation, and apoptosis [34]. It has been demonstrated that many helminths secrete calreticulin with the ability to inhibit host immune responses to facilitate the survival of the parasite within host. Among them, some helminth-expressed CRTs were able to bind to some complement components [19,35]. In our previous studies, we have identified that *T. spiralis*-produced CRT (*Ts*-CRT) could effectively bind to C1q to inhibit the C1q-initiated complement classical activation [9]. More specifically, the binding region of CRT to C1q was located within the S domain, which spans the N and P domains of *Ts*-CRT [10].

Recently, we identified that *E. multilocularis* alveolar larvae-derived CRT conferred the similar binding ability to human complement component C1q and inhibited the activation of the classical pathway of complement, as well as other non-complement-activated immune functions in which C1q is involved [24]. As a target recognition molecule of the lectin pathway, MBL is highly similar in structure and functions to C1q that initiates classical pathway [36]. A previous study confirmed that the diagnostic antigen C antigen on the surface of *E. multilocularis* contains a polysaccharide component, mainly galactose and N-acetylglucosamine residues [37]. These results indicate *Em*CRT is possibly involved in the inhibition of MBL-mannose-initiated lectin pathway of complement activation in addition to the C1q-involved classical pathway. We thus further explored the interaction of *Em*CRT with MBL and its role in the potential inhibition of lectin pathway of complement activation as an additional approach to evade host complement attack. In this study, we indeed demonstrated the binding of *Em*CRT to exogenous MBL (Figure 2) and to physical MBL in serum (Figure 5) as well, suggesting that *Em*CRT is able to inhibit the lectin pathway of complement through binding to MBL. Structural analysis and molecular docking of *Em*CRT with human MBL also exhibited the interaction and binding capacity between MBL and *Em*CRT (Figure 3). Further experiment confirmed that the binding of r*Em*CRT to external MBL or natural MBL in serum indeed inhibited the MBL-mannose-initiated lectin pathway leading to the decreased formation of complement intermediate products C3b and C4b (Figure 6).

It is a common strategy for parasites or other pathogens to produce molecules to inhibit lectin complement activation pathway. Dengue virus produced a nonstructural NS1 that is able to bind to MBL to inhibit its recognition by immune cells and the formation of the membrane-attacking complex on its membrane [38]. *Candida tropicalis* secreted asparagine protease 1 (SAPS1) to cleave MBL and inhibit the activation of the lectin pathway [39]. *Klebsiella pneumoniae* camouflaged its capsule components to avoid recognition by MBL and ficolins [40]. Similarly, flatworm *Fasciola hepatica* secreted a serine protease inhibitor (*Fh*Srp) that inhibited MBL binding to its surface and the subsequent MBL-mediated activation of the lectin pathway [41]. *Sarcoptes scabiei* secreted SMIIPP-Ss D1 and I1, members of the protein-hydrolyzing inactive serine protease paralogues (SMIIPP-Ss) family, bound to the MBL and inhibited lectin pathway [42].

Our previous study has identified the C1q-binding site on *Ts*-CRT as located within the 154 amino acids of the S domain, between the N and P domains [10]. Considering the structural similarity between MBL and C1q [36], it is possible that *Em*CRT shares the similar binding site to C1q and MBL as well. To determine the actual MBL binding region on *Em*CRT, different fragments of *Em*CRT including *Em*CRT-N, *Em*CRT-P, *Em*CRT-S, *Em*CRT-C, and *Em*CRT-NP were expressed as corresponding recombinant proteins. The binding assays with different *Em*CRT fragments confirmed that r*Em*CRT-S possessed the similar MBL binding capacity to that of full-length r*Em*CRT and r*Em*CRT-NP, significantly higher than those conferred by other fragments (Figure 4). The MBL binding site within S-domain is consistent to the molecular docking analysis with strong interaction prediction between *Em*CRT and MBL A-chain within the amino acids between N domain and P domain of *Em*CRT (Figure 3B). The evidences of actual binding assays between *Em*CRT fragments and MBL and MBL-mannose-initiated lectin pathway assays demonstrated that *Em*CRT-S possesses the strongest binding capacity to MBL and highest inhibitory ability to lectin activation pathway (Figure 3, Figure 4, Figure 5 and Figure 6), further confirming the MBL binding region is located in S-domain of *Em*CRT.

All results revealed in this study demonstrated that *E. multilocularis* calreticulin was able to bind to MBL so as to block MBL-mannose-mediated lectin complement activation pathway. The MBL binding site on *Em*CRT is located within the S-domain of *Em*CRT. Combining with the inhibition of C1q-initiated classical complement activation identified in our previous study and the inhibition of MBL-mannose-initiated lectin pathway revealed in this study, we determine that *Em*CRT plays an important role in the immune evasion of *E. multilocularis* alveolar larvae against host complement attack as a survival strategy within human tissue. Thus, *Em*CRT could be a good candidate for vaccine and drug development against *E. multilocularis* infection [24,27]. This is supported by our previous study showing immunization with r*Em*CRT-induced protective immunity against infection of *E. multilocularis* in BALB/c mice. The immunological mechanism underlying the *Em*CRT-induced protective immunity and its potential therapeutic effect on established infection is under investigation.

## Figures and Tables

**Figure 1 pathogens-14-00354-f001:**
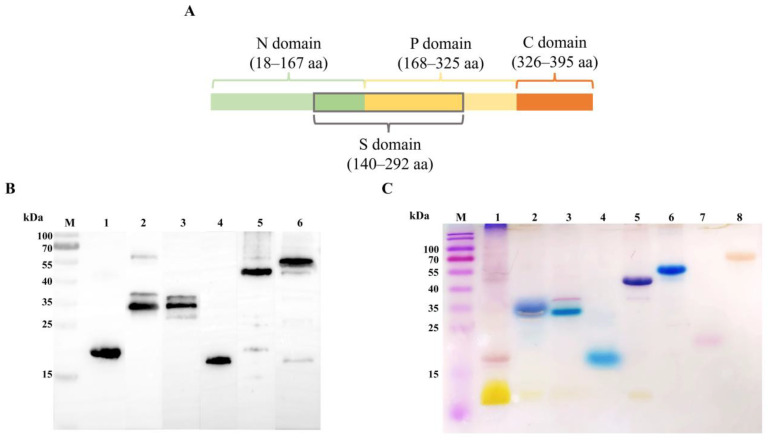
*Em*CRT fragmentation and calcium binding ability. (**A**) Schematic diagram of *Em*CRT fragmentation of each functional domain. (**B**) Western blot showing 1 μg of r*Em*CRT and its fragments including r*Em*CRT-N (Line 1), r*Em*CRT-P (Line 2), r*Em*CRT-S (Line 3), r*Em*CRT-C (Line 4), r*Em*CRT-NP (Line 5), and r*Em*CRT (Line 6) recognized by moused anti-His monoclonal antibody. (**C**) Purified r*Em*CRT and its fragments (5 μg each) were stained with Stains-all in blue, indicating their calcium binding ability: r*Em*CRT-P (Line 2), r*Em*CRT-S (Line 3), r*Em*CRT-C (Line 4), r*Em*CRT-NP (Line 5), and r*Em*CRT (Line 6) stained in blue, while r*Em*CRT-N (Line 1), r*Em*Fer (Line 7) as a *E. multilocularis* irrelevant protein control, and BSA (Line 8) control were stained in red, indicating these proteins have no calcium-binding activity. (Original Western blot and SDS-PAGE images of subfigures (**B**,**C**) can be found in Appendix A.)

**Figure 2 pathogens-14-00354-f002:**
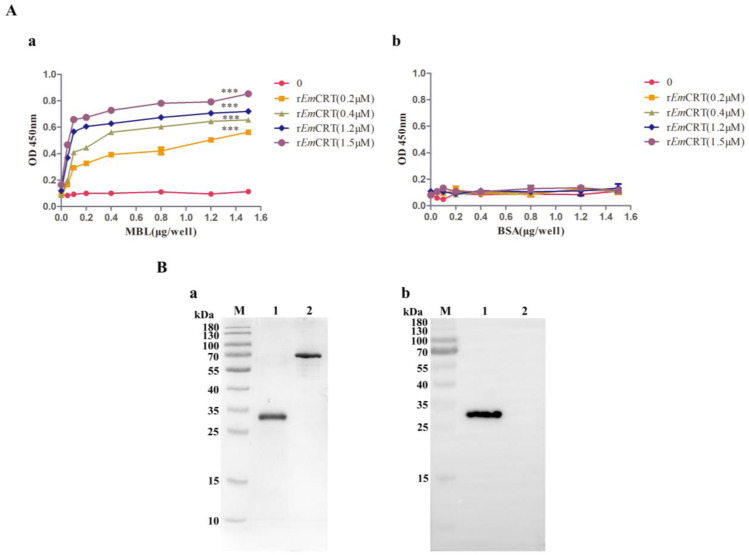
r*Em*CRT binding to human MBL. (**A**) ELISA results showing r*Em*CRT (0, 0.2, 0.4, 1.2, 1.5 μM/well) was able to bind onto MBL coated on the plates (0, 0.5, 1, 2, 4, 8, 12, 15 μg/mL) detected by anti-His monoclonal antibody. (**a**) Nothing bounds to BSA coated on the plates (0, 0.5, 1, 2, 4, 8, 12, 15 μg/mL). (**b**) Data are expressed as mean of OD_450_ ± SDs of three independent experiments (*** *p* < 0.0005 compared to BSA-coated plates). (**B**) Far western blot showing r*Em*CRT was able to bind onto MBL transferred on PVDF membrane. Separation of 1 μg MBL by SDS-PAGE under denaturing condition (**a**). Far western blot with anti-His antibody (1:1000) (**b**). Lane 1: 1 μg MBL. Lane 2: 1 μg BSA. (Original Western blot and SDS-PAGE images of subfigures (**B**) can be found in Appendix A).

**Figure 3 pathogens-14-00354-f003:**
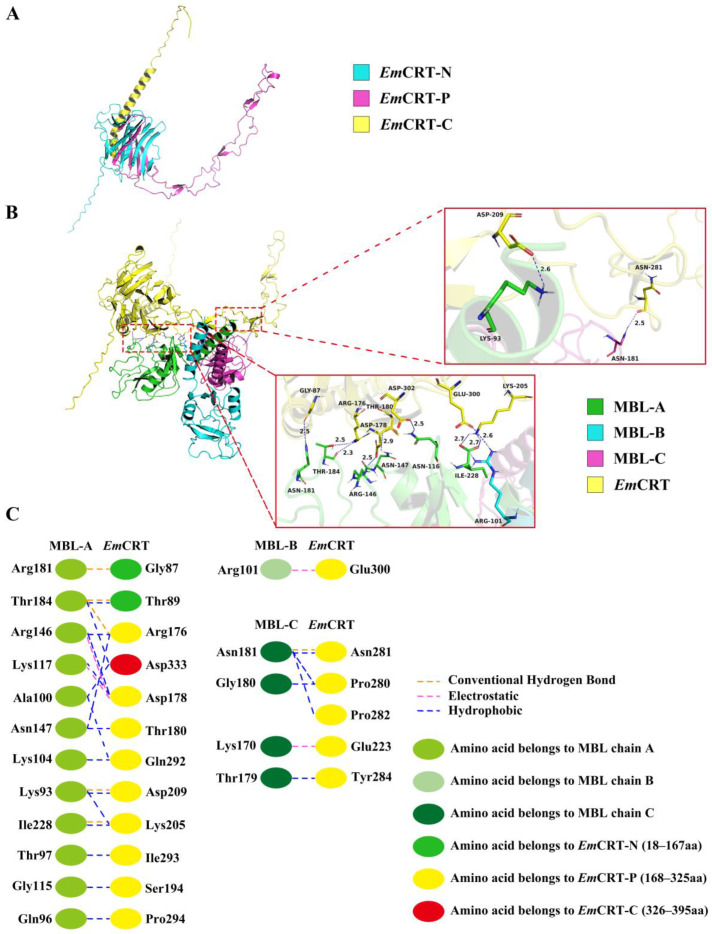
Predicted protein–protein molecular docking of *Em*CRT with MBL (PDB:1HUP) 3D model. (**A**) 3D model obtained by homology modelling of *Em*CRT shows the N domain (cyan), the P domain (rosy red), the C domain (yellow). (**B**) Predicted MBL binding regions on *Em*CRT are located in the N domain and P domain with detail of interacting amino acids between *Em*CRT and MBL showing on the right panel. (**C**) Interface analysis of *Em*CRT and MBL binding regions showing the interacting amino acids and their forces of action.

**Figure 4 pathogens-14-00354-f004:**
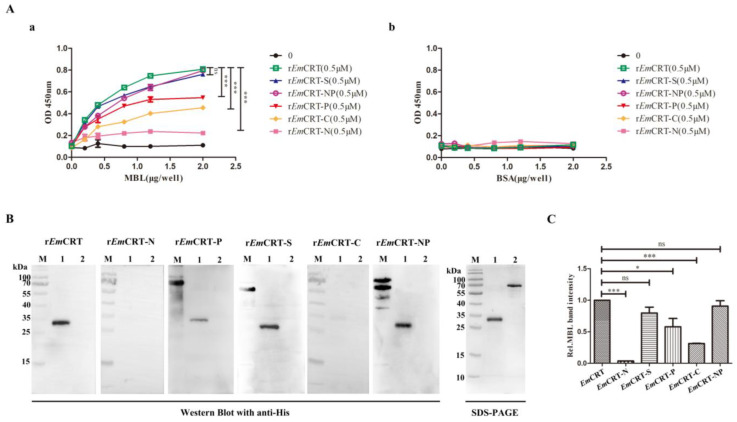
Binding of recombinant proteins of *Em*CRT and its fragments to MBL. (**A**) ELISA showing the binding of r*Em*CRT and the expressed recombinant fragments (0.5 μM/well) to MBL coated on the plates (0, 2, 4, 8, 12, 20 μg/mL) (**a**) but not binding to BSA coated on plates (0, 2, 4, 8, 12, 20 μg/mL). (**b**) The bound recombinant proteins were detected by anti-His monoclonal antibody. Data are expressed as mean of OD_450_ ± SDs of three independent experiments (*** *p* < 0.0005, ns, not significantly different compared to plates coated with BSA). (**B**) Far western blot was performed to detect r*Em*CRT and its fragments (15 μg/mL) bound onto MBL (1 μg) transferred on membrane, visualized by anti-His antibody (1:1000). SDS-PAGE of MBL and control BSA was shown on the right panel. (**C**) Densitometry reading of recognized bands. The binding ability of r*Em*CRT-S and r*Em*CRT-NP was similar to r*Em*CRT. (* *p* < 0.005, *** *p* < 0.0005, ns, no significant difference compared to bands incubated with r*Em*CRT). (Original Western blot and SDS-PAGE images of subfigures (**B**) can be found in Appendix A).

**Figure 5 pathogens-14-00354-f005:**
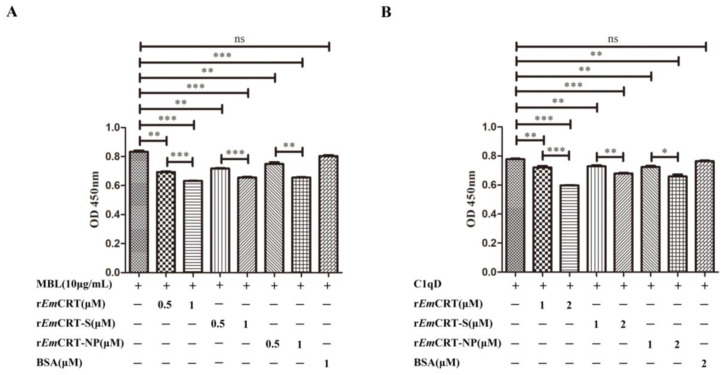
r*Em*CRT and its MBL binding regions inhibited binding of MBL to mannan. (**A**) Pre-incubation of different amounts of r*Em*CRT, r*Em*CRT-S, and r*Em*CRT-NP (0, 0.5, 1.0 µM) with MBL (10 μg/mL) before adding to plates coated with mannan (50 μg/mL), detected with anti-MBL monoclonal antibody. (**B**) Under the same ELISA condition, different amounts of r*Em*CRT, r*Em*CRT-S, or r*Em*CRT-NP (0, 1, 2 μM) were pre-incubated with C1qD (1:50) before adding into plates coated with mannan. Data are expressed as mean of OD_450_ ± SDs of three independent experiments (* *p* < 0.005; ** *p* < 0.001, *** *p* < 0.0005, ns, not significantly different compared to the reading from MBL without incubation with recombinant *Em*CRT).

**Figure 6 pathogens-14-00354-f006:**
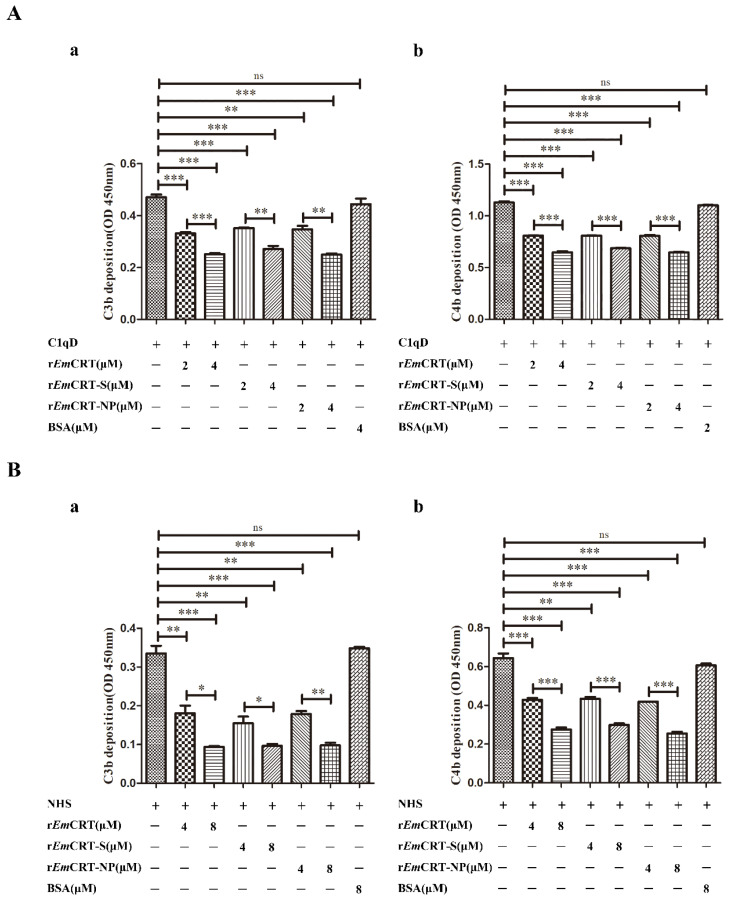
Inhibition of C3b and C4b deposition by r*Em*CRT and its functional binding regions measured by ELISA. (**A**) Pre-incubation of 100 μL C1qD (1:50) as source of natural MBL with different amounts of r*Em*CRT, r*Em*CRT-S, or r*Em*CRT-NP (0, 2, 4 μM) before adding into mannan-coated plates (50 μg/mL). After being washed, C1qD (1:150) was added as source of complement components to initiate lectin pathway of complement activation. Deposition of C3b (**a**) and C4b (**b**) was detected with anti-C3 or C4 monoclonal antibodies. (**B**) The similar ELISA procedure was performed using NHS as source of physiological MBL instead of C1qD serum to detect the generation of C3b (a) and C4b (b). Data are expressed as mean of OD_450_ ± SDs of three independent experiments (* *p* < 0.005, ** *p* < 0.001, *** *p* < 0.0005, ns, no significant difference compared to plates without added r*Em*CRT).

## Data Availability

Data included in this article are available from the corresponding author upon request.

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
