# Peer review of "Echinococcus multilocularis* Calreticulin Inhibits Lectin Pathway of Complement Activation by Directly Binding to Mannose-Binding Lectin"

_pathogens, 2025, doi:10.3390/pathogens14040354_

Round 1

Reviewer 1 Report

Comments and Suggestions for Authors

General comments

The manuscript presented by Yuxiao Shao et al. is quite interesting, as exploring the immune evasion mechanisms developed by E. multilocularis,  they demonstrate the role of calreticulin secreted by the parasite can inhibit the activation of the MBL complement pathway.  

The numerous strategies used by parasites to escape the host immune system to survive are not fully understood, and this experimental study brings relevant new insights into this issue as well as could promote potential therapeutical applications. 

Overall, the MS is well written, the methodology and experiments applied seems adequate; the results obtained are quite robust for the purpose. However, as I am not an expert on the field of 3D structure analysis, other comments should be done. 

Author Response

Point 1: Overall, the MS is well written, the methodology and experiments applied seems adequate; the results obtained are quite robust for the purpose. However, as I am not an expert on the field of 3D structure analysis, other comments should be done.

Response: Thank you for your positive comments. We deeply appreciate your recognition of our research work. The 3D structural prediction of EmCRT 3D was performed using AlphaFold3. Docking of human MBL (PDB: 1HUP) to the EmCRT model was performed using the protein-protein docking tool HDOCK.

Reviewer 2 Report

Comments and Suggestions for Authors

The work aims to study the function of calreticulim in echinococcosis. The search for new molecules and candidates for vaccines and new drugs is extremely relevant. The work is well written, with a clear methodology and good results that can support future studies.

In the methodology, I suggest including in the statistics section whether the number of five samples was representative of the sample. We have statistical tests to assess the sample size for the study. And it was also not clear whether the sample underwent any sample distribution test. It is important to add this information.

Author Response

Point 1: In the methodology, I suggest including in the statistics section whether the number of five samples was representative of the sample. We have statistical tests to assess the sample size for the study. And it was also not clear whether the sample underwent any sample distribution test. It is important to add this information.

Response: Thank you for your constructive suggestion and comments. Regarding the sample distribution, we analyzed the data for normal distribution through the Kolmogorov-Smirnov test. All experiments were performed in vitro without animal test, and repeated at least once. All experimental samples were duplicated. The statistical analysis of the data was carried out using the two-tailed Mann–Whitney test by GraphPad Prism 5 software. P< 0.05 was considered as statistically significant difference. The detail statistical analysis methods have been added into revised Methods.